

**Measurement Report:**
**Size distributions of inorganic and organic components in particulate matter from**
**a megacity in northern China: dependence upon seasons and pollution levels**
**Yingze Tian**[*,a]**, Yinchang Feng**[a]**, Yongli Liang**[a]**, Yixuan Li**[a]**, Qianqian Xue**[a]**, Zongbo**
**Shi**[b]**, Jingsha Xu**[b]**, Roy M. Harrison**[*,b†]
[a] State Environmental Protection Key Laboratory of Urban Ambient Air Particulate Matter Pollution
Prevention and Control, College of Environmental Science and Engineering, Nankai University,
Tianjin, 300071, China
[b] School of Geography Earth and Environmental Science, University of Birmingham, Birmingham,
B15 2TT, UK
Correspondence to: Roy M. Harrison (r.m.harrison@bham.ac.uk), Yingze Tian
(tianyingze@hotmail.com)
[†]Also at: Department of Environmental Sciences / Center of Excellence in Environmental Studies, King
Abdulaziz University, PO Box 80203, Jeddah, 21589, Saudi Arabia





**Abstract:**
Size distributions of inorganic and organic components in particulate matter (PM)
provide critical information on its sources, fate and pollution processes. Here, ions,
elements, carbon fractions, n-alkanes, polycyclic aromatic hydrocarbons (PAHs),
hopanes and steranes in size-resolved (9 stages) PM were analyzed during one year in
a typical northern Chinese industrial megacity (Tianjin). We found that the
concentrations of organic carbon fraction OC3, $NO_3^-$ (or $SO_4^{2-}$) and the sum of crustal
elements were the highest in the pseudo-ultrafine (<0.43 μm), fine (0.41-2.1 μm) and
coarse (>2.1 μm) modes, respectively. The diagnostic ratios of organic components
consistently suggest that the traffic influence was stronger during summer and coal
combustion during winter. Nitrate and high molecular weight PAHs were concentrated
in the fine mode during winter, while nitrate and low molecular weight PAHs showed
bimodal distributions especially during summer due to repartitioning. Long-chain *n*-
alkanes showed a peak in the coarse mode during spring and summer, indicating a
relatively stronger vegetation source and resuspended dust. Furthermore, we found a
major difference in the size distribution of aerosol components during heavy pollution
episodes ($PM_{10}$>233 μg m$^{-3}$) in different seasons: in spring, OC fractions, 4- and 5-ring
PAHs, hopanes and C18-C33 n-alkanes were enhanced at 1.1-3.3 μm, implying that
they may arise from local combustion sources which emit relatively large particles; in
summer PM mass, $SO_4^{2-}$, $NH_4^+$, Al, and C26-C33 *n*-alkanes were enhanced mainly in
the coarse mode, peaking at 5.8-9.0 μm, indicating a large contribution from
resuspended dust or heterogeneous reactions on dusts; in the winter and autumn, $NO_3^-$



was significantly enhanced followed by $SO_4^{2-}$, $NH_4^+$, OC and EC with their peaks
shifting from 0.43-0.65 µm to 0.65-2.1 µm, indicating strong atmospheric processing.
These results reveal that the size distributions of inorganic and organic aerosol
components are dependent on the seasons and pollution levels as a result of the differing
sources and physicochemical processes.


**1 Introduction**


Atmospheric particulate matter (PM) negatively affects human health and visibility,
influences global climate change and nutrient cycles within ecosystems (Burnett et al.,
2014; Wang et al., 2014; Zhang et al., 2017). Properties and effects of PM depend on
their chemical composition and particle sizes (Seinfeld and Pandis, 1998; Kanakidou et
al., 2005; Kompalli et al., 2020). PM cover a wide range of sizes from a few nanometers
to several hundreds of micrometers, and is composed of a complex mixture of inorganic
substances (such as water-soluble ions, elements, and elemental carbon) and hundreds
of organic compounds. Several studies have shown that the size distribution of chemical
components can provide evidence for examining the sources and formation pathways
(Guo et al., 2014; Yao et al., 2018; Hilario et al., 2020).
Non-polar organic compounds in PM, such as n-alkanes, polycyclic aromatic
hydrocarbons (PAHs), hopanes and steranes, can provide specific information to
identify PM sources (Oros and Simoneit, 2000; Wang et al., 2009; Han et al., 2018).
PAHs are mainly emitted from anthropogenic activities, such as biomass burning, coal



combustion, oil combustion and industrial processes (Mastral et al., 1996; Liu et al.,
2012). Hopanes and steranes are abundant in coal, crude oils and the lubricant oil
fraction, and are often found in traffic exhaust and coal combustion emissions (Oros
and Simoneit, 2000). n-Alkanes can arise from both natural and anthropogenic activities,
including abrasion products from vegetation leaf surfaces (characterized by the
predominance of > C29 odd n-alkanes) and fossil fuel combustion (Han et al., 2018).
Although these organic compounds are assumed to be relatively stable and nonreactive
(Feng et al., 2006; Ma et al., 2011), they can undergo complex physical and chemical
changes, like photochemical oxidation and gas-particle partitioning (Robinson et al.,
2006; Han et al., 2018). The size distributions of the organic compounds are strongly
associated with their sources, physical behavior and chemical reactions. Several studies
on size distributions of PM chemical components have been conducted for ions,
elements, and carbon fractions (Dillner et al., 2006; Huang et al., 2016; Tian et al., 2016;
Hilario et al., 2020), and for some organic compounds (Wang et al., 2009; Han et al.,
2018; Xu et al., 2020). These studies indicated that the size distributions of the inorganic
or organic components may vary with season and degree of pollution, so characterizing
the variations is valuable for understanding their sources and fate. However, the
combined analysis of seasonal variations of inorganic and organic components based
on whole year sampling has rarely been conducted.
Severe atmospheric PM pollution has been a recurring problem affecting developing
countries (Cheng et al., 2016; Zou et al., 2017). As one of the world's fastest-developing
economic zones, the Jing-Jin-Ji economic circle in China has experienced severe and





long-lasting haze episodes (Guo et al., 2014; Long et al., 2019). The "Jin" is Tianjin, a
megacity with a population of 15.6 million. With gross industrial production ranking
third among all Chinese provinces (TMSB, 2017), Tianjin is a typical industrial city.
Various anthropogenic emissions, including industrial emissions, coal combustion,
vehicle exhaust and resuspended dust, significantly contribute in this region.
Additionally, the chemical and physical characteristics of PM are complex during
different seasons. High reactivity due to relative humidity (RH) during winter has been
reported (Cheng et al., 2016), and strong dust and sea salt emissions during spring and
summer can increase the surfaces needed for heterogeneous reactions. The factors that
influence heavy pollution in different seasons remain unclear due to their complexity
(Cheng et al., 2016; Yao et al., 2018). The variation of size-resolved inorganic and
organic chemical component size distributions during periods of heavy pollution in
different seasons can throw light on the mechanisms of PM pollution (Tian et al., 2016;
Xu et al., 2020).
In this study, we conducted a comprehensive study of water-soluble ions, elements,
carbon fractions, n-alkanes, PAHs, hopanes and steranes in size-resolved PM (9 stages)
samples collected from May 2018 to April 2019 in Tianjin, China. The goal of this study
is to (1) determine size-resolved chemical composition, including inorganic and organic
components, (2) characterize seasonal variations of size distribution of inorganic and
organic markers, (3) explore the mechanisms of PM pollution during different seasons
through investigating the changes in composition as well as size distribution. The
results of this work will help researchers better understand the sources, physical process



and chemical mechanisms of PM pollution during different seasons, which is crucial
for evaluating the effects of PM on human health, visibility and regional radiative
forcing, and for developing control strategies.

**2 Methods and materials**
**2.1 Study area and sampling**
Tianjin is located at 116°43'E~118°04'E, 38°34'N~40°15'N, in the northern part of
the North China Plain. It covers an area of 11947 km$^2$, near the Capital of China (Beijing)
and bordering on the Bohai Sea. The population is 15.6 million. As a municipality
directly under the Central Government and one of the first coastal open cities, Tianjin
is a typical industrial city and a major manufacturing base for some industrial products,
whose gross industrial production ranks third among all Chinese provinces (TMSB,
2017). The industries include petroleum and chemical, modern metallurgy, aerospace,
automobile and equipment manufacturing, amongst others. More than 20 million tons
of coal is consumed by industries and residential heating every year. The civil motor
vehicle fleet in Tianjin is about 2.8 million. The Tianjin Port is the largest
comprehensive port in northern China and a major port for foreign trade.
The PM samples were collected by an Andersen air sampler (Andersen Series E-0162,
USA) with 9-stage size ranges of >9.0, 9.0-5.8, 5.8-4.7, 4.7-3.3, 3.3-2.1, 2.1-1.1, 1.1-
0.65, 0.65-0.43, and < 0.43 μm. Quartz-fibre filters (81 mm in diameter) were used.
The sampling period was from May 2018 to April 2019, involving spring (May in 2018,
and March and April in 2019), summer (June and August in 2018), autumn (1





September to 15 November 2018) and winter (15 November 2018 to 15 March 2019,
when coal was consumed for residential heating). The sampling was conducted for 47
h during spring and summer, and for 23 h during autumn and winter, and was stopped
during rainy days. Overall, 153 sets of successfully size-resolved samples (total 1377
fraction samples) were obtained. Additionally, the $PM_{2.5}$ and $PM_{10}$ mass concentrations
were continuously monitored by a Beta Particulate Monitor (BPM-200, Focused
Photonics, China) at the sampling site. The correlation plots between $PM_{2.1}$ and $PM_{10}$
mass concentrations sampled by the Andersen sampler (PM-measured) vs
corresponding means of continuous $PM_{2.5}$ and $PM_{10}$ concentrations monitored by the
BPM (PM-monitored) were showed in Figure S1. High correlations (0.86 and 0.82)
were observed.

To explore the mechanisms of heavy pollution, the heavy pollution was defined as

the days with $PM_{10}$ concentrations higher than the 3rd quartile of all the samples, which
was 233 $\mu g\ m^{-3}$.

**2.2 Chemical analysis of ions, elements and carbon fractions**

In this work, 7 ions, 17 elements, 7 carbon fractions, 18 PAHs, 2 cholestane, 7 hopane

and 24 n-alkanes were analyzed on each size range. For ion analyses, including $Na^+$,
$K^+$, $Ca^{2+}$, $NH_4^+$, $Cl^-$, $NO_3^-$ and $SO_4^{\ 2-}$, one eighth of each quartz filter was cut. The
sample was placed in a centrifuge tube which had been ultrasonically cleaned and dried,
before addition of 8 ml of distilled deionized water, then placed in an ultrasonic bath
for 20 minutes. The sonicated centrifuge tube was refrigerated for 24 hours. The clear



liquid in the middle of the centrifuge tube was withdrawn with a needle tube, filtered
through two 0.2 μm filters, and injected into the sample bottle. Lastly the extraction
liquid was analyzed to determine the cation and anion concentrations with a Thermo
ICS900 Ion Chromatograph (Thermo Electron). For the elemental analysis, 17 elements
(Al, Ca, Ti, K, Mg, Na, V, Fe, Cu, Zn, Mn, Pb, As, Cd, Co, Cr, Ni) were analyzed by
inductively coupled plasma-mass spectrometry (ICP-AES) (IRIS Intrepid II, Thermo
Electron). We cut one eighth of each quartz filter into portions and extracted elements
into acid solution ($HNO_3$: HCl: $H_2O_2$ = 1: 3: 1) using a microwave digester (PyNN
Corporation). Carbon fractions were analyzed by a thermal/optical carbon aerosol
analyzer (DRI 2001A, Atmoslytic Inc.), which gives OC1, OC2, OC3, OC4, OP, EC1,
EC2, EC3 fractions. The instrument is based on heating and releasing organic carbon
and elemental carbon at different temperatures, and uses He-Ne laser to allow
quantification of the pyrolytic (OP) fraction. The IMPROVE-A thermal/optical
reflectance (TOR) protocol was used (Chow et al., 1993) with 140°, 280°, 480°, 580°,
740°, and 840°C to divide the carbon fractions.

**2.3 Chemical analysis of organic components**
Organic compounds in half of total sample filters were analyzed by gas
chromatography-mass spectrometry (GC-MS). The full names and corresponding
abbreviations of organic compounds were summarized in Table 1.

**Table 1**

For extraction, the filters were cut into portions and put into tubes. Then 10 ml of



dichloromethane (DCM) and 10 ml of n-hexane (1:1, v:v) were added into each tube
which was then put in an ultrasonic bath at 30 °C for 15 minutes. The solution was
filtered with a silica column to clean up the extract. The silica column was eluted with
20 ml of DCM/hexane (1:1) and the extract was collected. After reduction to less than
5 ml (about 2 or 3 ml) by rotary evaporator, the extract was solvent-exchanged to n-
hexane. Finally, the volume was reduced to 1 ml. Calibrations of PAHs, hopanes,
steranes and n-alkanes were used to test chromatographic conditions before analysing
samples. A DB-5MS fused-silica capillary column (30 m × 0.25 mm, 0.25 mm film
thickness, Agilent Technology) was used in the GC separation procedure. The carrier
gas was pure helium (purity of 99.99% or more) at a constant flow rate of 1.0 mL min$^{-}$
$^{1}$. For PAHs, hopanes and steranes, inlet and transfer line temperatures were set to 230 °C
and 280 °C respectively. EI mode was used and the ionization energy level was 70eV.
For n-alkane analysis, inlet and transfer line temperatures were set to 300 °C.

**2.4 Quality assurance and quality control (QA/QC)**
Throughout the whole process, much attention was given to ensure quality assurance
and control. All samples were collected by one instrument and analyzed by the same
methods. When it rained, we stopped the sampling campaign. The air flow rate was
corrected by a flowmeter before each sampling period. Before use, the quartz filters
were baked in an oven at 400-500 °C to eliminate any organic matter that may exist on
the filters. All the filters were equilibrated at room temperature for 48 h in desiccators
before weighting. Each filter was weighted by a sensitive microbalance with balance





sensitivity ±0.010 mg. The quartz filters were kept in aluminum foil bags before and
after sampling until analysis, and stored at −4 °C. The samples were analyzed within
30 days.

Field and laboratory blanks were measured to correct the corresponding data.

Standard reference materials were analyzed with the same procedure every day and the
recovered values for all the target components showed low relative standard deviations.
Additionally,, the first sample of every ten samples was re-examined with the precision
found to be within 10%. The recoveries were 79-106% for elements, and 96-110% for
ions. For carbon fractions, a system stability test (three-peak detection) is required
before and after detecting samples and the relative standard deviation should not exceed
5%. For organic compounds, all extractions were conducted two times so that the
samples were extracted adequately, with the tubes sealed by foil and ice added to the
ultrasonic bath. The internal standards (naphthalene-d8, acenaphthene-d10,
phenanthrene-d10, chrysene-d12, and perylene-d12; hexamethylbenzene; n-
tetracosane-d50) were used for the samples to qualify actual volumes of the target
compounds present. The calibration curves were strongly linear. The recoveries of most
organic compounds ranged from 70%-130%, except for Nap, Any, Ana, C10, and C11
with recoveries below 50%.

**2.5 Diagnostic ratios**

Diagnostic ratios were used to identify the sources. The Ant/(Ant+Phe) ratio for

PAHs is used to differentiate petroleum origins or pyrogenic sources, and IPY/BghiP





is for different fossil fuels. The ratio of C29αβ/C30αβ for hopane can also be used to
judge fossil fuels. The homohopane index C34[S/(S+R)] can distinguish fuel maturity,
which is defined as:
$$C34[S/(S+R)] = \frac{C34\alpha\beta S}{C34\alpha\beta S + C34\alpha\beta R}$$   (1)
For n-alkanes, the carbon preference index (CPI) can reflect the comparison between
natural and anthropogenic contributions, which is defined as the ratio of the total
concentration of odd n-alkanes to that of even n-alkanes:
$$CPI = \frac{\sum_{i=5}^{16} C_{2i+1}}{\sum_{i=5}^{16} C_{2i}}$$   (2)
where i is the carbon number. Due to that plant wax n-alkanes show strong odd carbon
number predominance, biogenic n-alkanes should have CPI values greater than unity,
whereas anthropogenic n-alkanes should have CPI values close to unity (Han et al.,

2018).

Contributions from natural wax n-alkanes (WNA%) and petrogenic n-alkanes
(PNA%) can directly present the origins. The negative value of $[C_i - (C_{i-1} + C_{i+1})/2]$
should be replaced by zero.
$$WNA\% = \frac{\sum_{i=10}^{32}[C_i - (C_{i-1} + C_{i+1})/2]}{\sum_{n=10}^{33} C_n}$$   (3)
PNA%=100%- WNA%   (4)

**3 Results and discussion**
**3.1 Size distribution of PM mass concentrations**
Figure 1 describes the size distribution of PM mass concentrations in spring, summer,
autumn and winter. The average concentrations of PM were the greatest during winter,

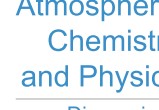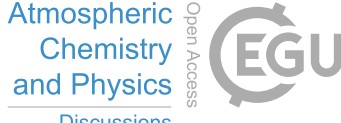

followed by spring, autumn and summer at most sizes. The size distribution of the PM
mass concentrations was bimodal, with one peak at 0.43-0.65 µm (fine mode) and the
other at 4.7-5.8 µm (coarse mode). Particles at small sizes may be mainly influenced
by secondary formation and anthropogenic sources (such as vehicle emissions, coal
combustion, industrial emissions, etc.), while particles at large sizes may be more
associated with natural sources, resuspended dust (such as road dust, construction dust,
mechanical abrasion processes, etc.) and industrial emissions (Hilario et al., 2020). In
this work, the peaks in the coarse mode were very strong during spring due to the
influences of resuspended dust and natural sources. The enhancement of PM in the fine
mode during winter is associated with unfavorable meteorological conditions for the
dispersion of fine particles, and increased emissions from coal combustion for heating
(Cheng et al., 2016; Tian et al., 2016). The broader peak in the fine mode during winter
should be noted.
**Figure 1**
To explore the causes of heavy pollution, size distributions of PM during the heavy
pollution days were compared with less polluted days (all the rest of the days).
According to the temporal variations of the size-segregated particle concentrations in
Figure S1, most heavy pollution cases occurred in winter and there were only a few
days in other seasons. Figure 1 also describes the size distribution of particle mass
concentrations for less polluted days and heavy pollution days during four seasons. PM
concentrations in the fine mode increased during heavy pollution in all of the seasons.
It was also found in several previous studies that fine particles significantly





accumulated during the haze pollution period (Wang et al., 2014; Tian et al., 2016). In
spring, the peak in the coarse mode was weaker during heavy pollution than during less
polluted days, while the concentrations in the coarse mode significantly increased
during summer heavy pollution. Thus, the spring heavy pollution may be associated
with sources or processes that engender small particles (like combustion, secondary
formation, etc.), while summer heavy pollution was strongly linked with large particle
sources (like resuspended dust, dust storms, etc.). During winter and autumn, it is
interesting to find that peaks in the fine mode shifted to a larger size, from 0.43-0.65
μm during less polluted days to 0.65-1.1 μm during heavy pollution. This result is
consistent with the previous studies, which showed that the peak mass concentration of
fine mode particles shifted to larger sizes during heavily polluted days (Tian et al., 2016;
Guo et al., 2014). The causes and mechanisms of heavy pollution will be further
explored according to the variations of inorganic and organic components.

**3.2 Size-resolved chemical compositions and diagnostic ratios**
Correlations among size-segregated chemical composition (percentages of 4 ions, 17
elements and 7 carbon fractions accounting for PM concentrations) have been
calculated and are summarized in Table S1(a). It is noticeable to find that the
compositions were similar in each mode (size<0.43 μm considered as the pseudo-
ultrafine mode; 0.43-0.65, 0.65-1.1 and 1.1-2.1 μm as the fine mode; and 2.1-3.3 3.3-
4.7, 4.7-5.8, 5.8-9.0 and >9.0 μm in the coarse mode), while they showed relatively
high differences between different modes. Additionally, the correlations among size-



segregated composition of organic compounds (percentages of 18 PAHs, 2 cholestane,
7 hopane and 24 n-alkanes accounting for PM concentrations) showed similar trends
(Table S1(b)), indicating that both main component composition and organic
composition were similar in each mode. Thus, the sizes were aggregated into three
modes to investigate the size-segregated compositions: coarse (>2.1 μm), fine (0.43-
2.1 μm), and pseudo-ultrafine (< 0.43 μm) modes.
**3.2.1 Size-segregated main species**
Concentrations of chemical species in the pseudo-ultrafine, fine and coarse modes
during the four seasons are shown in Figure 2. The composition of main species were
similar during spring, autumn and winter: in the pseudo-ultrafine mode, the primary
component was organic carbon fraction OC3 (2.4, 2.0 and 3.2 μg m$^{-3}$ during spring,
autumn and winter, respectively); in the fine mode, the primary component was $NO_3^-$
(8.0, 12.7 and 15.1 μg m$^{-3}$, respectively); and in the coarse mode, the CE (crustal
elements, defined as sum of Al, Ca, Fe and Ti) was highest with concentrations of 17.2,
16.0 and 20.2 μg m$^{-3}$. During summer, the highest components were $SO_4^{2-}$ in the
pseudo-ultrafine (1.6 μg m$^{-3}$) and fine modes (8.2 μg m$^{-3}$), and CE in the coarse mode
(14.8 μg m$^{-3}$). The sums of OC fractions were higher than other components in most
modes and seasons. $NO_3^-$, $SO_4^{2-}$ and OC are mainly from secondary formation,
combustion sources and industrial emissions, and crustal elements are linked with
resuspended dust (except during dust storm periods).
The concentration of $SO_4^{2-}$ is the highest among all measured species in the fine
mode during summer, while $NO_3^-$ was the primary component during other seasons.



During summer, photochemical processes can be more efficient. However, due to the
thermodynamic instability of ammonium nitrate, it may decompose under high
temperature (Hasheminassab et al., 2014). It is interesting to find that the $NO_3^-$
concentration in the coarse mode was higher during summer (5.7 μg m$^{-3}$) than during
other seasons, which may result from the deposition of nitric acid vapour upon coarse
particle surfaces.

**Figure 2**

**3.2.2 Size-segregated organic compounds and diagnostic ratios**

PAHs arise mainly from anthropogenic sources. According to the concentrations of

organic compounds in Figure 2, the concentrations of $\sum_{18}$PAHs (summed
concentrations of 18 PAHs) were 22, 19, 31 and 49 ng m$^{-3}$ during spring, summer,
autumn and winter, respectively. The 5 and 6-ring PAHs (mainly BbF, BeP, BaP, DBA,
IPY and BghiP) showed higher concentrations than other PAHs during all four seasons
(Figure S2a). High molecular weight PAHs are emitted under a high temperature
condition as from vehicles and they are more stable in the particle phase. It is interesting
to observe an obvious increase of 4-ring PAHs in the fine mode during winter, which
may be caused by coal combustion for residential heating, and the low wintertime
temperature favouring partition into the condensed phase. Tianjin is a typical city in
northern China with considerable coal burning for house heating during winter, which
emits more 4-ring PAHs (Zhang et al., 2017). Moreover, some low-molecular-weight
PAHs can be emitted by volatilization and low to moderate temperature combustion,
such as during coking (Khalili et al., 1995), biomass combustion (Zhang et al., 2008)





and residential utilization of electricity or gas for heating and cooking (Yadav et al.,
2018). Thus, the results suggest that vehicle exhaust, coal combustion, industrial
emission and biomass burning had mixed effects on the atmospheric PAH pollution,
and that coal burning was one of major sources of PAHs in northern China, especially
during winter.
Additionally, diagnostic ratios of organic compounds are may be indicative of
sources, so the ratios in size-resolved PM during each season and their ranges for
sources are summarized in Table 2. The Ant/(Ant+Phe) ratios were greater than 0.1,
confirming an influence from pyrogenic emissions (Han et al., 2018). The IPY/BghiP
ranged from 0.66 to 2.51, indicating mixed effects of vehicle emissions and coal
combustion (Grimmer et al., 1983). The IPY/BghiP values were lower in the pseudo-
ultrafine fraction, implying that vehicle emissions impact was stronger in the pseudo-
ultrafine than in other modes. Enhancement of IPY/BghiP ratios during autumn and
winter indicates that PAHs derived from coal combustion increased, consistent with the
results of 4-ring PAH variations. Due to the low concentrations of organic compounds,
the diagnostic ratios are subject to uncertainty, especially for the seasons and modes
with lower concentrations.
**Table 2**
Hopanes and steranes are often used to determine the fuel maturity (Oros and
Simoneit, 2000). As shown in Figure 2, the total concentrations of 2 steranes and 7
hopanes ranged from 1 to 5 ng m$^{-3}$ for the pseudo-ultrafine mode, from 4 to 25 ng m$^{-3}$
for fine mode and from 3 to 27 ng m$^{-3}$ for the coarse mode. The maximum occurred in



winter and the minimum occurred in summer. The $17\alpha(H),21\beta(H)$-30-norhopane
(C29$\alpha\beta$) and $17\alpha(H),21\beta(H)$-30-hopane (C30$\alpha\beta$) showed the highest levels of the
hopanes and steranes during four seasons in all modes, as shown in Figure S2b. The
C30$\alpha\beta$ mass concentration was higher than C29$\alpha\beta$ during spring, summer and autumn,
while the contrary occurred during winter. C34$\alpha\beta$S and C34$\alpha\beta$R concentrations were
significantly enhanced during autumn and winter in the fine and coarse modes.
The molecular composition of hopanes are further assessed by their mass ratios. As
shown in Table 2, diagnostic ratios of C29$\alpha\beta$/C30$\alpha\beta$ indicate a strong traffic influence
during summer (C29$\alpha\beta$/C30$\alpha\beta$=0.59-0.69) and a strong coal combustion contribution
during winter (ratio=1.32-1.38). The hopane C34[S/(S+R)] ratio, which is an indicator
of the maturity of combusted fossil fuel, ranged from 0.27 to 0.75. Hopane [S/(S+R)]
ratios increase with increasing fuel maturity. When the level of the R isomer is much
higher than the concentration of the S isomer, PM is mainly influenced by emissions
from coal combustion, while the similar concentrations of R and S isomer indicate
traffic emissions. It was reported that the homohopane index [S/(S+R)] for coal smoke
samples increases with coal rank (lignite 0.05; brown coal 0.09; sub-bituminous coal
0.20; bituminous coal 0.35) (Oros and Simoneit, 2000). The C34[S/(S+R)] ratios in this
study were more indicative of coal burning during autumn and winter. And we notice
that the C34[S/(S+R)] values were generally in the order of pseudo-ultrafine mode >
fine mode > coarse mode, indicating that the maturity of the fossil fuel decreased with
the increase of particle size. Obvious low values of C34[S/(S+R)] in the coarse mode
during winter may indicate an influence of combustion of immature coals.



n-Alkanes are mainly emitted in vehicle emissions, coal combustion, tire-wear
particles and particulate abrasion products from leaf epicuticular waxes, and it has been
reported that they have unique signatures for different sources (Han et al., 2018). N-
alkanes were the most abundant organic compounds, with the total concentrations
ranging from 2187 during summer to 4452 ng m$^{-3}$ during winter (Figure 2). As shown
in Figure S2 and Table 2, the values of the carbon number of the most abundant n-
alkane (Cmax) in this study were C23 in the fine mode and C31 in the coarse mode
during spring and summer, and were generally C28 in all modes during autumn and
winter. Cmax represents the carbon number of the most abundant n-alkane, which is
regarded as an important indicator of biogenic inputs. C23 has been reported as
indicative of vehicle emissions (Lyu et al., 2017). C31 mainly comes from a vegetation
source (plant wax) and tire-wear source which can be caused by resuspended dust (Han
et al., 2018). Thus, the Cmax values in this work suggest a strong fuel combustion
influence in the fine mode, and contributions of vegetation sources and resuspended
dust in the coarse mode during spring and summer. During autumn and winter, the
Cmax=28 was considered to come from local emissions, especially coal combustion
(Han et al., 2018).
The CPI (carbon preference index) ranged from 0.73 to 1.48, suggesting a
predominant contribution from anthropogenic sources (Han et al., 2018). WNA% and
PNA% provide a direct insight that above 75% of n-alkanes were originated from
anthropogenic sources. CPI and WNA% consistently demonstrate that n-alkanes were
mainly contributed by anthropogenic activities, and the natural source was higher in the





coarse mode during spring and summer, consistent with the interpretation of other ratios
above.

**3.3 Seasonality of size distributions**
**3.3.1 Size distributions of main components**
The size distributions of main chemical species (ions, elements and carbon fractions)
during spring, summer, autumn and winter are shown in Figure 3a. $NO_3^-$, $SO_4^{2-}$ and
$NH_4^+$ mass concentrations were abundant in the fine mode, which exhibited maxima at
0.43-0.65 or 0.65-1.1 µm during most seasons, except for $SO_4^{2-}$ in winter (1.1-2.1 µm)
and $NO_3^-$ in summer (3.3-4.7 µm). Except for primary sources, the fine mode $NO_3^-$ and
$SO_4^{2-}$ can be formed from the gaseous and aqueous phase reactions, and the coarse
mode $NO_3^-$ and $NH_4^+$ can origin from repartitioning and heterogeneous reactions with
sea salt and crustal dust (Liu et al., 2017). A strong peak of $NO_3^-$ was observed at 3.3-
5.8 µm during summer. Due to the thermodynamic instability, fine mode $NH_4NO_3$ can
be dissociated into vapour at high ambient temperatures and then shifted onto coarse
particles by the condensation, reaction, dissolution or coagulation. During spring and
summer, the weak peaks of $NH_4^+$ in the coarse mode may partly derive from suspended
soil containing fertilizer. The size distributions of $Cl^-$ showed obvious seasonal
variations, which strongly peaked in the fine mode during winter and autumn
corresponding to emissions of HCl from combustion sources which can form semi-
volatile ammonium chloride (Pio and Harrison, 1987); and peaked in the coarse mode
during summer because of sea salt.


**Figure 3**
Carbon fractions generally exhibited typical bimodal distributions with peaks in fine
and coarse modes. The levels of OC2, OC4 and EC1 in the fine mode and those of OC3
in both fine and coarse modes were significantly enhanced during winter. The OC
fractions and EC1 primarily come from coal combustion, vehicle exhaust, and biomass
burning, while EC2 and EC3 mainly originate from diesel and oil combustion (Kim and
Hopke, 2004; Shi et al., 2016). Thus, the enhancement of these carbon fractions in the
fine mode during winter is strongly linked with coal combustion for residential heating.
The OC3 in the coarse mode may be influenced by complex emissions and reactions. It
has been shown that PM emissions in northern China are complex during winter (Tian
et al., 2016), like scattered (area source) coal combustion. Shi et al. (2020) indicated
that scattered coal is the largest source of ambient volatile organic compounds during
the heating season in Beijing, which are important precursors for secondary organic
carbon (SOC). Thus, SOC generation may be another reason for OC enhancement in
both fine and coarse modes. Photochemical reactions may be generally weak during
winter, but high precursor concentrations, high humidity and high PM concentrations
during winter may enhance the aqueous phase and heterogeneous reactions.
The crustal elements (Al, Ca, Fe, Ti, etc.) were primarily concentrated in the coarse
mode at 4.7-5.8 or 5.8-9.0 μm. The high wind speed during spring can facilitate the
resuspension of dust (such as road dust, soil dust) into the atmosphere and result in the
high level of the crustal elements in the coarse mode. K showed a bimodal distribution:
the peak in the fine mode may be associated with biomass or coal combustion, and that



in coarse mode associated with natural sources from soil dust and sea salt. Figure S3
shows the ratios between water-soluble ions and corresponding elements in each size.
Peaks of $K^+/K$ and $Ca^{2+}/Ca$ were generally in the fine mode, indicating that K and Ca
may be contributed by industrial or combustion sources in the fine mode and strongly
associated with crustal dust in the coarse mode. A strong peak of $Na^+/Na$ in the coarse
mode during summer may indicate impacts of sea salt, consistent in explaining the peak
of $Cl^-$ in the coarse mode during summer.
**3.3.2 Size distribution of organic markers**
Figure 3b describes the size distribution of organic compounds during different
seasons. The size distribution varied with seasons and congeners. About 66-79% PAHs
were in the pseudo-ultrafine and fine modes, which were higher than the cumulative
percentages of hopanes (64-71%) and n-alkanes (56-60%). The sum of 2- and 3-ring
PAHs showed a bimodal distribution, while the summed concentrations of 6- and 7-
ring PAHs was strongly concentrated in the fine mode. High molecular weight PAHs
are less volatile and predominantly formed on smaller particles where they condense
immediately after combustion. However, low molecular weight PAHs are more volatile,
so they are easily adsorbed on larger particles as the emissions cool down, or can
evaporate from the particle-phase into the air and subsequently adsorb/condense onto
pre-existing coarser particle (Offenberg and Baker, 1999). The 4- and 5-ring PAHs were
found with a unimodal distribution peaking in the fine mode (0.43-1.1 µm) during
winter, while they were bimodal during other seasons, due to their repartitioning being
enhanced under higher ambient temperatures.



Most sterane and hopanes detected in this work were observed with a bimodal
distribution. It is interesting to observe the enhancement of 17β(H),21β(H)-hopane
(C30ββ) and 17α(H),21β(H)-22R-tetrakishomohopane (C34αβR) in the coarse mode
during winter. C30ββ and C34αβR have been regarded as markers of less mature coals,
such as lignite and sub-bituminous coal combustion (Oros and Simoneit, 2000). Thus,
their increase indicates a contribution from immature coal combustion to coarser PM
during winter. For most n-alkanes, a bimodal size distribution was found at 0.43-0.65
or 0.65-1.1μm in the fine mode and 3.3-47 or 4.7-5.8 μm in the coarse mode. Total
concentrations of C26-C33 showed a strong peak in the coarse mode during spring and
summer, indicating a relatively stronger vegetation source and a resuspended dust
contribution to PM in the coarse mode.
In previous publications, a bimodal distribution of low molecular weight PAHs and
a strong peak in the fine mode of high molecular weight PAHs were reported (Lv et al.,
2016; Han et al., 2018). The bimodal distribution of most n-alkanes was observed by
Wang et al. (2009), Lyu et al. (2017) and Xu et al. (2017). The size distributions of
hopanes were reported as unimodal in the fine mode (Kleeman et al., 2008; Han et al.,
2018), and can be bimodal during summer (Wang et al., 2009). Through comparing
these publications with this work, we find that the size distributions of most organic
compounds were consistent, but the peaks in the coarse mode are often stronger in this
work. The difference can be due to variations in their sources, increased fine particle
coagulation at high particle concentrations, and organic compound repartitioning.
Tianjin is an important heavily populated megacity in northern China, which has large





emissions from industry and traffic, and high resuspended dust due to strong human
activities (such as construction and heavy trucks activity). The stronger peaks in the
coarse mode, especially during spring and summer, may be linked with the high
resuspension of dust from coal combustion, industrial emissions, traffic emissions,
construction dust, and mechanical abrasion processes.

**3.4 Size distributions during less polluted and heavy pollution periods**

To explore which sizes and which components are significantly enhanced during

heavy pollution periods, the enhancement ratios, which are defined as ratios of
component concentration during heavy pollution days to that during other days
($ER_{H/C}=C_H/C_C$), are calculated and shown in Figure 4. There are different formation
mechanisms for heavy pollution episodes during different seasons. During spring heavy
pollution days, the $ER_{H/C}$ of OC3 and OC4 at 1.1-3.3 μm were significantly high, along
with the enhancement of 4- and 5-ring PAHs (Flt, Pry, BbF, BkF, Bap), hopane and
C18-C33 at these sizes. Further, Figure 5 compares the size distributions of species
which were significantly enhanced during heavy pollution. During the spring heavy
pollution, the peaks of the OC3 shifted to 2.1-4.7 μm, and the peaks of 4-ring PAHs,
C29αβ-HP, C30αβ-HP and sum of C26-C33 also changed to a larger size. The results
imply that this heavy pollution may be caused by local combustion sources which
emitted relatively larger particles.

Figure 4

Figure 5



The summer heavy pollution was characterized by strong enhancement of $SO_4^{2-}$,
$NH_4^+$, Al, EC2 and C26-33 mainly in the coarse mode (Figure 4). The peak of Al in the
coarse mode changed from 4.7-5.8 μm for less/average polluted days to 5.8-9.0 μm for
heavy pollution (Figure 5). And $SO_4^{2-}$, $NH_4^+$ and C26-33 increased in size to 3.3-4.7 and
5.8-9.0 μm respectively. Consistent with the discussion on size distribution of PM mass
concentrations, this heavy pollution may be caused by resuspended dust or long-range
transported dust and heterogeneous reactions. Suspended soil containing fertilizer can
contribute to alkaline conditions and enhance the heterogeneous reactions on coarse
particles (Shen et al., 2011). Additionally, the increases of the low molecular weight
PAHs and short chain n-alkanes demonstrate the repartitioning and suspension.
Heavy pollution during autumn and winter was strongly associated with significant
enhancement of $NO_3^-$ and moderate enhancement of $SO_4^{2-}$, $NH_4^+$, OC and EC at most
sizes especially at 0.65-3.3 μm, as shown in Figure 4. $NO_3^-$, $SO_4^{2-}$, $NH_4^+$, and OC can
be from chemical reactions and partly from primary emissions, while EC is from
primary emissions. The enhanced EC demonstrates that the meteorological conditions
during heavy pollution caused accumulation of particles and precursors. Referring to
Figure 5, peaks of $NO_3^-$, $SO_4^{2-}$, OC3 and EC1 in the fine mode shifted from 0.43-0.65
μm during less polluted days to 0.65-1.1 or 1.1-2.1 μm during heavy pollution. Heavy
pollution during autumn and winter in northern China is usually characterized by large
emissions and specific meteorological condition with a stable boundary layer, weak
winds, an increase in temperature, and high relative humidity (RH) (Cheng et al., 2016).
The meteorological condition was unfavorable for the dispersion of particles and





precursors and favours secondary particle formation, hygroscopic growth and stronger
coagulation. Lower temperature, high RH and high precursors increase the formation
of ammonium nitrate by facilitating the aqueous phase chemical reactions which was
more likely to occur at 0.65-2.1 μm (Zhang et al., 2013; Tian et al., 2016). In addition,
as discussed above, the emissions during winter were more complex and emitted
coarser EC-containing particles (such as domestic coal combustion), which can rapidly
accumulate during heavy pollution conditions. The EC at a coarser size can strength the
reaction at these sizes, because EC could provide sites for adsorption and reaction due
to its large surface area, and it has the catalytic properties for redox chemistry reactions.
Stronger relationship between black carbon (BC)-containing particles and secondary
species during more polluted periods were also observed by Wang et al. (2019).

**4. Summary and conclusions**
A comprehensive study on water-soluble ions, elements, carbon fractions, n-alkanes,
PAHs, hopanes and steranes in size-resolved PM samples were conducted during 1 year
in a typical northern Chinese industrial megacity. Size distributions of the inorganic and
organic components during different seasons and pollution levels were analyzed.
The size distribution of the PM mass concentrations was bimodal during all four
seasons peaking at 0.43-0.65 μm and 4.7-5.8 μm; and the coarse mode peak was large
during spring and the fine mode peak was more substantial during winter. Both main
component composition and organic composition were similar within each mode, but
relatively different for the different modes. The OC3, $NO_3^-$ and sum of crustal elements





showed the highest concentrations in the pseudo-ultrafine, fine and coarse modes,
respectively, except that $SO_4^{2-}$ became the largest component in the pseudo-ultrafine
and fine modes during summer. For organic markers, PAHs, C29αβ/C30αβ and
C34[S/S+R] ratios consistently indicate stronger traffic influence during summer and
increased coal combustion during winter; and imply that the maturity of the fossil fuel
source decreased with the increase of particle size. The enhancement of C30ββ and
C34αβR in the coarse mode during winter indicate a contribution from immature coal
combustion. The profile of *n*-alkanes suggests a dominant fuel combustion influence in
the fine mode, and contributions of a vegetation source and resuspended dust in the
coarse mode especially during spring and summer.
For the size distributions, $NO_3^-$, $SO_4^{2-}$ and $NH_4^+$ concentrations were large in the fine
mode during most seasons, while $SO_4^{2-}$ peaked at 1.1-2.1 µm during winter probably
due to a large contribution of aqueous phase reactions, and $NO_3^-$ peaked at 3.3-4.7 µm
during summer due to repartitioning. Carbon fractions generally exhibited typical
bimodal distributions. The crustal elements (Al, Ca, Fe, Ti, etc.) were primarily
concentrated in the coarse mode at 4.7-5.8 or 5.8-9.0 µm. Most sterane, hopane and n-
alkanes were observed to have a bimodal distribution. High molecular weight PAHs
were concentrated at small sizes during winter, while low molecular weight PAHs were
frequently bimodal due to repartitioning.
During the spring heavy pollution periods, OC3, OC4, 4- and 5-ring PAHs, hopane
and C18-C33 n-alkanes were enhanced at 1.1-3.3 µm, and their peaks shifted to a larger
diameter, implying that the heavy pollution may be caused by a local combustion source





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





**Figure captions**

**Figure 1** Size distribution of particle mass concentrations during spring, summer, autumn and winter for seasonal average, less polluted samples and heavy pollution samples (with PM concentrations higher than the 3rd quartile).

**Figure 2** Concentrations of chemical species in the pseudo-ultrafine mode (size<0.43 μm), fine mode (0.43-2.1μm) and coarse mode (>2.1μm) during four seasons.
CE is crustal elements, defined as sum of Al, Ca, Fe and Ti.
Full names and abbreviations of organic compounds are listed in Table 1.

**Figure 3 a** Size distribution of main species during spring, summer, autumn and winter.
**b** Size distribution of organic compounds during spring, summer, autumn and winter.

**Figure 4** Concentration enhancement ratios (which is defined as ratios of component concentration during heavy pollution days to that during less polluted days, $ER_{H/C}=C_H/C_C$) of main species and organic compounds in each size during four seasons.

**Figure 5** Size distribution of main species during heavy pollution and less polluted days for spring, summer, autumn and winter.





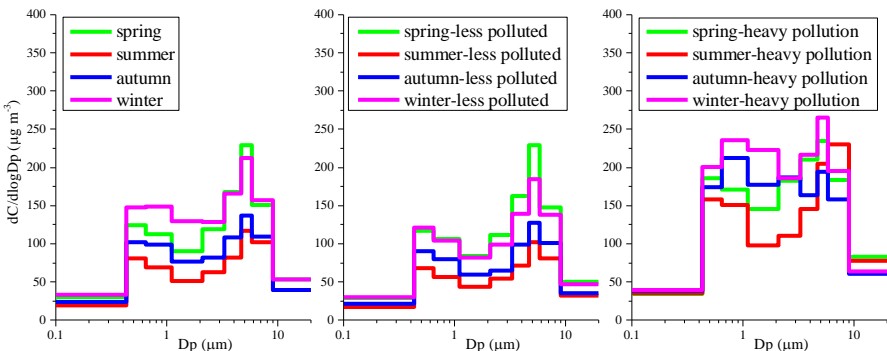

763

**Figure 1** Size distribution of particle mass concentrations during spring, summer, autumn and winter for seasonal average, less polluted samples and heavy pollution samples (with PM concentrations higher than the 3rd quartile).

767



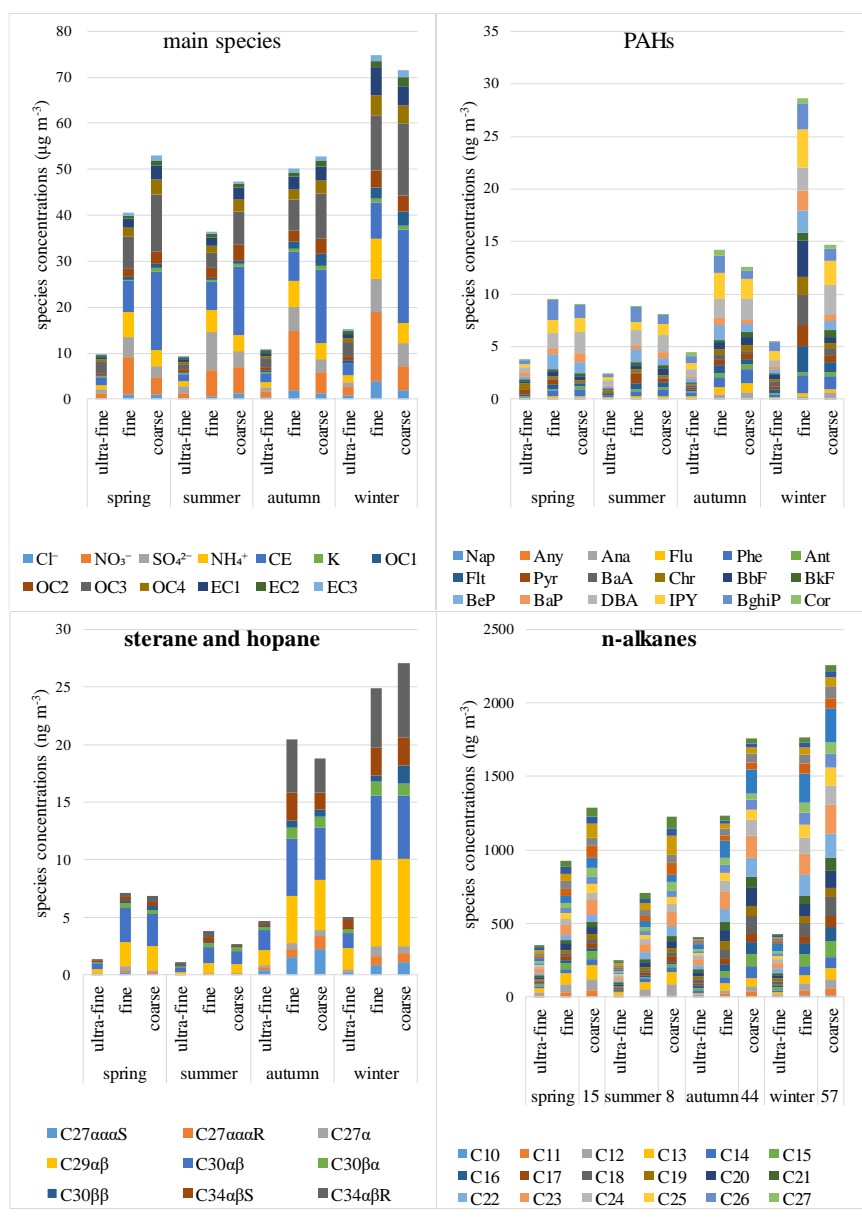

**Figure 2** Concentrations of chemical species in the pseudo-ultrafine mode (size<0.43 μm), fine mode (0.43-2.1μm) and coarse mode (>2.1μm) during four seasons.

CE is crustal elements, defined as sum of Al, Ca, Fe and Ti.

Full names and abbreviations of organic compounds are listed in Table 1.

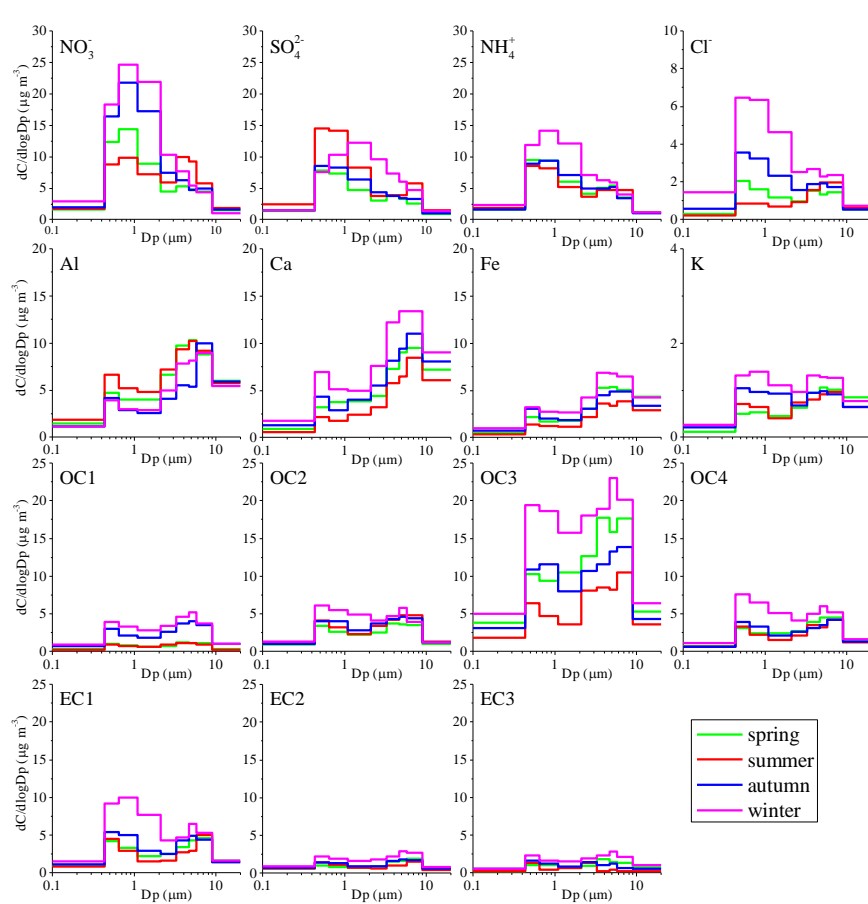

774

**Figure 3a** Size distribution of main species during spring, summer, autumn and winter.





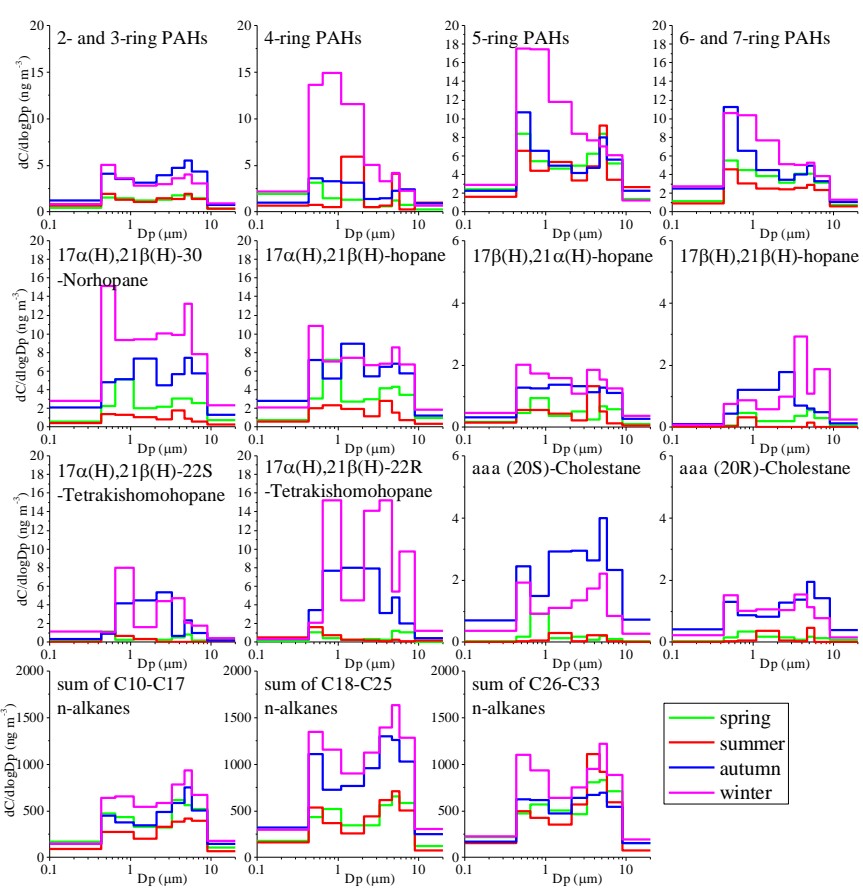

**Figure 3b** Size distribution of organic compounds during spring, summer, autumn and winter.



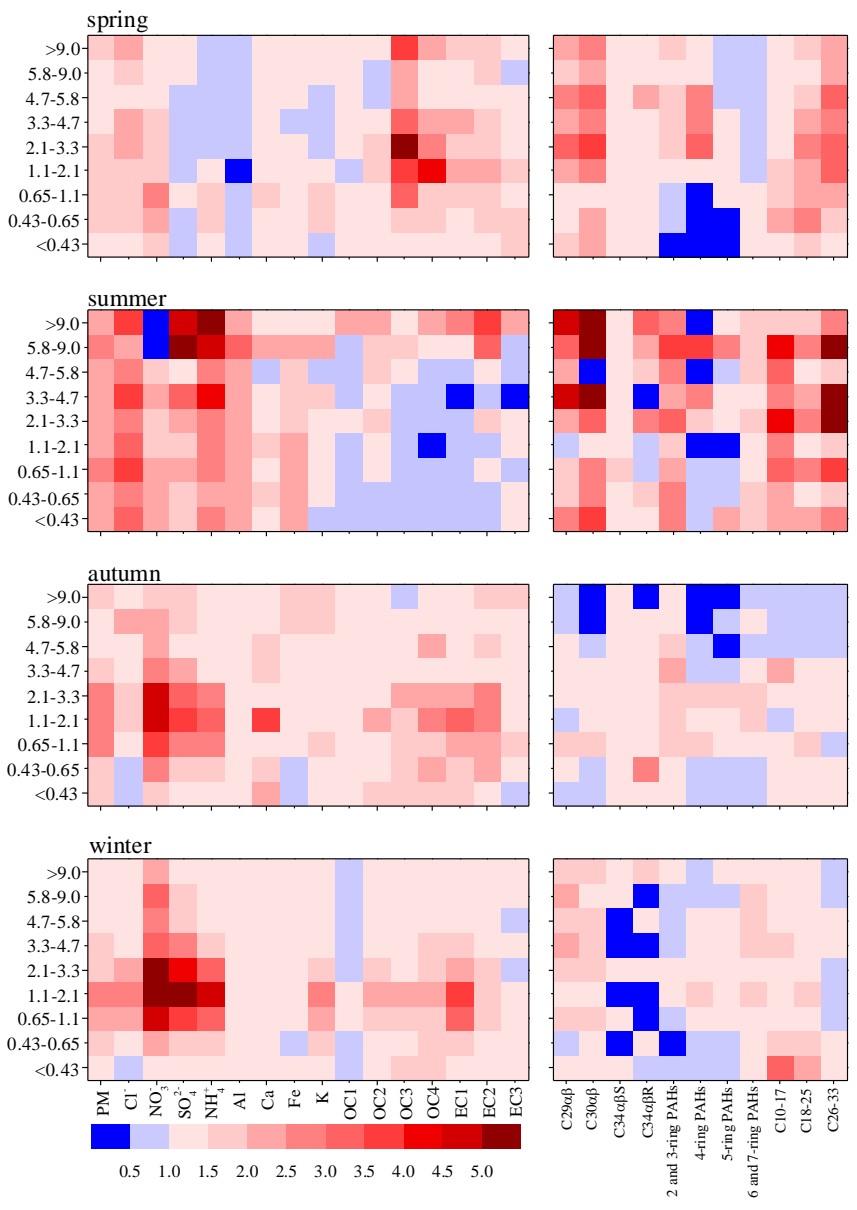

**Figure 4** Concentration enhancement ratios (which is defined as ratios of component concentration during heavy pollution days to that during less polluted days, $ER_{H/C}=C_H/C_C$) of main species and organic compounds in each size during four seasons.






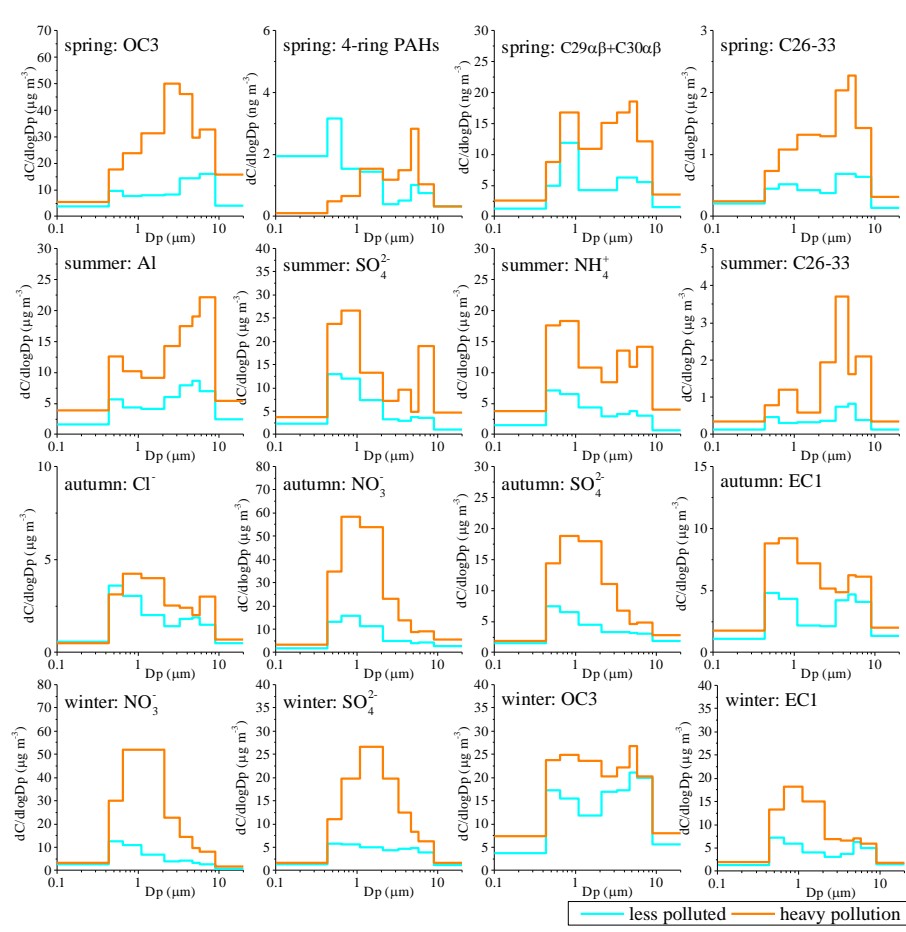


**Figure 5** Size distribution of main species during heavy pollution and less polluted days for spring, summer, autumn and winter.






**Table 1** Full names and abbreviations of organic compounds.

| Full name | Abbreviation | Full name | Abbreviation |
|---|---|---|---|
| **PAHs** | | **n-alkanes** | |
| Naphthalene | Nap | n-Decane | C10 |
| Acenaphthylene | Any | Undecane | C11 |
| Acenaphthene | Ana | Dodecane | C12 |
| Fluorene | Flu | Dridecane | C13 |
| Phenanthrene | Phe | Tetradecane | C14 |
| Anthracene | Ant | Pentadecane | C15 |
| Fluoranthene | Flt | Hexadecane | C16 |
| Pyrene | Pyr | Heptadecane | C17 |
| Benz[a]anthracene | BaA | Octadecane | C18 |
| Chrysene | Chr | Nonadecane | C19 |
| benzo[b]fluoranthene | BbF | Icosane | C20 |
| Benzo[k]fluoranthene | BkF | Henicosane | C21 |
| Benzo[e]pyrene | BeP | Docosane | C22 |
| Benzo[a]pyrene | BaP | Tricosane | C23 |
| Dibenzo[a,h]anthracene | DBA | Tetracosane | C24 |
| Indeo[1,2,3-cd]pyrene | IPY | Pentacosane | C25 |
| Benzo[g,h,i]perylene | BghiP | Hexacosane | C26 |
| Coronene | Cor | Heptacosane | C27 |
| **Sterane and hopane** | | Octacosane | C28 |
| $\alpha\alpha\alpha$ (20S)-Cholestane | C27$\alpha\alpha\alpha$S | Nonacosane | C29 |
| $\alpha\alpha\alpha$(20R)-Cholestane | C27$\alpha\alpha\alpha$R | Triacontane | C30 |
| 17a(H)-22,29,30-Trisnorhopane | C27$\alpha$ | Hentriacontane | C31 |
| 17$\alpha$(H),21$\beta$(H)-30-Norhopane | C29$\alpha\beta$ | Dotriacontane | C32 |
| 17$\alpha$(H),21$\beta$(H)-hopane | C30$\alpha\beta$ | Tritriacontane | C33 |
| 17$\beta$(H),21$\alpha$(H)-hopane | C30$\beta\alpha$ | | |
| 17$\beta$(H),21$\beta$(H)-hopane | C30$\beta\beta$ | | |
| 17$\alpha$(H),21$\beta$(H)-22S-Tetrakishomohopane | C34$\alpha\beta$S | | |
| 17$\alpha$(H),21$\beta$(H)-22R-Tetrakishomohopane | C34$\alpha\beta$R | | |




**Table 2** Diagnostic parameters and isomeric ratios of organic compounds in size-resolved PM during each season.

| spring | Ant/(Ant+Phe) | IPY/BghiP | C29αβ/C30αβ | C34[S/(S+R)] | Cmax | CPI | WNA% | PNA% |
|---|---|---|---|---|---|---|---|---|
| ultra | 0.24 | 0.70 | 0.78 | 0.62 | C23 | 1.33 | 19% | 81% |
| fine | 0.22 | 0.67 | 0.73 | 0.51 | C23 | 1.22 | 18% | 82% |
| coarse | 0.23 | 1.09 | 0.74 | 0.63 | C31 | 1.64 | 24% | 76% |
| summer | | | | | | | | |
| ultra | 0.23 | 0.64 | 0.69 | 0.49 | C23 | 0.97 | 18% | 82% |
| fine | 0.27 | 0.57 | 0.59 | 0.41 | C23 | 1.10 | 17% | 83% |
| coarse | 0.28 | 1.17 | 0.68 | 0.35 | C31 | 1.41 | 24% | 76% |
| autumn | | | | | | | | |
| ultra | 0.27 | 0.75 | 0.73 | 0.48 | C23 | 0.81 | 18% | 82% |
| fine | 0.25 | 1.64 | 0.82 | 0.33 | C28 | 0.86 | 18% | 82% |
| coarse | 0.23 | 2.51 | 0.97 | 0.35 | C28 | 0.78 | 18% | 82% |
| winter | | | | | | | | |
| ultra | 0.24 | 0.87 | 1.32 | 0.75 | C28 | 0.76 | 18% | 82% |
| fine | 0.15 | 1.53 | 1.33 | 0.33 | C28 | 0.83 | 14% | 86% |
| coarse | 0.24 | 1.92 | 1.38 | 0.25 | C28 | 0.86 | 16% | 84% |
| values in references | > 0.1 for pyrogenic sources [a] | 0.2, 0.5 and 1.3 for gasoline, diesel and coal combustions [b] | 0.6-0.7 for gasoline, 0.4 for diesel [c], 0.6-2.0 for coal [d] | 0.05-0.35.for coal, larger for vehicles [d] | C31 for resuspended dust, C23 for vehicle | CPI < 2 petrogenic sources | | |

a Han et al., 2018
b Grimmer et al., 1983
c Rogge et al., 1993
d Oros and Simoneit, 2000