# Peer review of "Measurement Report"

_Atmospheric Chemistry and Physics, 2020_

## Referee Comment (RC1) · Anonymous Referee #1 · 10 Aug 2020

The authors conducted a comprehensive analysis of the size distributions of different chemical species in Tianjin. Size distributions of chemical compositions based on cascade impactor have been widely reported for over 20 years already, and there are many such studies in mega cities in China. The paper is a report of the results and it is difficult to see new contributions to the science of air pollution. I tried to identify something new in science and technology in the paper. The tools of the chemical analysis and the data analysis of the size distribution results are both very conventional. In the abstract, the authors said "These results reveal that the size distributions of inorganic and

organic aerosol components are dependent on the seasons and pollution levels as a result of the differing sources and physicochemical processes." While this is true, it is hardly exciting.

One possible way to upgrade the paper is to include modeling of the size distribution results. If this cannot be done, I suggest that the authors go to a lower tier journal to report the data. Not suitable for ACP at the moment.

---

## Referee Comment (RC2) · Anonymous Referee #2 · 17 Sep 2020

The study by Tian et al. reports an analysis of size segregated chemical composition of particulate matter in a megacity environment in China. The quality of analysis is excellent and the report is well written, however, there is nothing new which would advance the scientific knowledge of megacity pollution. There are at least one hundred published papers about PAHs and many hundreds of papers about particulate matter sources in that same Tianjin city region. Clearly, the topic is over researched and there must be well identified knowledge gaps to even plan a similar study. A general statement "The factors that influence heavy pollution in different seasons remain unclear

due to their complexity" does not warrant publication in ACP. The study would stand a much better chance if combined with planned source apportionment study (so perhaps the effort suffers from fragmentation), but even then it would hardly be exiting due to numerous other studies of the same topic. The category of Measurement Report paper in ACP should be reserved in my opinion to studies which are not presenting new findings, but for which geographical coverage is lacking. That is not the case with this study as neither methods, nor findings are new, nor the geographical coverage in that particular region is lacking. Given the quality and presentation of the paper it definitely deserves to be published, but in lower tier journals perhaps or as Data paper to keep credit for well executed analysis. Based on the above I have to recommend a rejection.

---

## Referee Comment (RC3) · Anonymous Referee #3 · 23 Sep 2020

The manuscript by Tien et al. reports an analysis of one year-long size-segregated chemical composition of particulate matter (PM) in Tianjin, a megacity in the northern China. The characterization is comprehensive and well conducted, the quality of the data is excellent and the manuscript is well written. Nevertheless, beyond the high quality of data and presentation, in my opinion the paper flaws of scientific relevance: no new analytical techniques or data analysis are used and it is hard to see new contributions to air pollution description or its scientific knowledge. This is especially problematic considering how many (hundreds?) studies exist reporting size distributions of PM chemical composition in Chinese mega cities (even in the same region). Sufficiently detailed examination of the existing literature on pollution in the region is also lacking as well as a careful description of the new results that this study aims to add to the scope. I don't personally have a clear idea of the requirements needed to publish in the "Measurement Report" category in ACP (this is an editorial choice more than a reviewer's task), but considering the journal level I think it should be reserved in any case to data presenting new findings, novel techniques and/or unique geographical coverage. After all, in the description of the Manuscript types on the ACP website it is clearly state that "Measurement reports present substantial new results", new results which are missing in the present study, in my opinion. All the above considerations prevent me from recommending publication in ACP in the present form. It may be a good idea (if the journal policy allows it) to give to the authors the opportunity to resubmit the paper with substantial scientific upgrades and/or to suggest the submission to lower tier journals.

---

## Author Comment (AC1) · 27 Oct 2020

Response: Thank you very much for the reviewers' time. Size distributions of chemical compositions have been reported in some literatures and demonstrated their importance in examining the sources and formation pathways. However, as far as we know, the size distributions of organic components are not so much, especially for their variations dependence upon seasons and pollution levels. Characterizing the variations is valuable for understanding their sources and fate. In addition, in the revision, we applied two-way and three-way factor analysis models to conduct size$-$resolved

source apportionment based on joint inorganic and organic components in the revision, so that the study can explore the better way to conduct size−resolved source apportionment. And more interesting findings were found. For example, two coal combustion associated factors were extracted. One Factor, which is characterized by high C30$\beta\beta$+C34$\alpha\beta$R, four−ring PAHs and carbon fractions at sizes 1.1−5.8 $\mu$m, was extracted because of its unique pattern of size distribution, and was identified as combustion of less mature coals. In addition, three factors associated with the formation processes of secondary components were identified. One factor, which was characterized by high NO3−, SO42− and NH4+ at sizes 1.1−3.3 $\mu$m, significantly correlated with relative humidity (RH) and is named as high RH-related secondary aerosol (RHSA). Peaks of PM, NO3- and SO42- shifted from the usual 0.43-0.65 $\mu$m to 1.1-2.1 $\mu$m during high RHSA periods, due to enhanced aqueous-phase reactions, hygroscopic growth and coagulation. This work demonstrates the value of size-resolved source apportionment of joint inorganic and organic markers in understanding the sources and physicochemical processing of PM.
* * *

---

## Author Comment (AC2) · 27 Oct 2020

Thank you very much. The size distributions of organic components are not so much, especially for their variations dependence upon seasons and pollution levels, which is valuable for understanding their sources and fate. In the revision, we would conduct the size-resolved source apportionment of particulate matter (PM) based on analyses of both inorganic and organic markers. While there have been many studies of particle source apportionment in China, this study differs in analyzing a measurement dataset which is both size fractionated and chemically speciated. There are very few published

studies of this type in the literature and our work shows major benefits of working with size-fractionated samples. The findings give valuable insights into both the sources of particles and the atmospheric processes affecting them.

---

## Author Comment (AC3) · 27 Oct 2020

Thank you very much. Size distributions of chemical compositions have been reported in some literatures and demonstrated their importance in examining the sources and formation pathways. However, as far as we know, the size distributions of organic components are limited. In the revision, size-resolved source apportionment of particulate matter (PM) was conducted by two-way and three-way receptor models, based on the one-year measurement of inorganic and organic tracers (n−alkanes, PAHs, hopanes and steranes). And more interesting findings were found.